# Solid-Phase “Self-Hydrolysis” of [Zn(NH_3_)_4_MoO_4_@2H_2_O] Involving Enclathrated Water—An Easy Route to a Layered Basic Ammonium Zinc Molybdate Coordination Polymer

**DOI:** 10.3390/molecules26134022

**Published:** 2021-06-30

**Authors:** Kende Attila Béres, István E. Sajó, György Lendvay, László Trif, Vladimir M. Petruševski, Berta Barta-Holló, László Korecz, Fernanda Paiva Franguelli, Krisztina László, Imre Miklós Szilágyi, László Kótai

**Affiliations:** 1Research Centre for Natural Sciences, Magyar Tudósok Krt 2, 1117 Budapest, Hungary; beres.kende.attila@ttk.hu (K.A.B.); lendvay.gyorgy@ttk.hu (G.L.); trif.laszlo@ttk.hu (L.T.); korecz.laszlo@ttk.hu (L.K.); fernandapaivafranguelli@edu.bme.hu (F.P.F.); 2Szentagothai Research Centre, Environmental Analytical and Geoanalytical Research Group, University of Pécs, Ifjúság Útja 20, 7624 Pécs, Hungary; istvan.sajo@gmail.com; 3Faculty of Natural Sciences and Mathematics, Ss. Cyryl and Methodius University, 1000 Skopje, North Macedonia; vladop@pmf.ukim.mk; 4Department of Chemistry, Biochemistry and Environmental Protection, Faculty of Sciences, University of Novi Sad, Trg Dositeja Obradovica 3, 21000 Novi Sad, Serbia; hberta@uns.ac.rswrit; 5Department of Inorganic and Analytical Chemistry, Budapest University of Technology and Economics, Műegyetem Rakpart 3, 1111 Budapest, Hungary; imre.szilagyi@mail.bme.hu; 6Department of Physical Chemistry and Materials Science, Budapest University of Technology and Economics, Műegyetem Rakpart 3, 1111 Budapest, Hungary; laszlo.krisztina@vbk.bme.hu; 7Deuton-X Ltd., Selmeci u. 89, 2030 Érd, Hungary

**Keywords:** ammine, hydrolysis, vibrational spectroscopy, thermal analysis, zinc molybdate, photocatalysis

## Abstract

An aerial humidity-induced solid-phase hydrolytic transformation of the [Zn(NH_3_)_4_]MoO_4_**@**2H_2_O (compound **1****@**2H_2_O) with the formation of [(NH_4_)*_x_*H_(1−*x*)_Zn(OH)(MoO_4_)]_n_ (x = 0.92–0.94) coordination polymer (formally NH_4_Zn(OH)MoO_4_, compound **2**) is described. Based on the isostructural relationship, the powder XRD indicates that the crystal lattice of compound **1@**2H_2_O contains a hydrogen-bonded network of tetraamminezinc (2+) and molybdate (2−) ions, and there are cavities (O_4_N_4_(μ-H_12_) cube) occupied by the two water molecules, which stabilize the crystal structure. Several observations indicate that the water molecules have no fixed positions in the lattice voids; instead, the cavity provides a neighborhood similar to those in clathrates. The **@** symbol in the notation is intended to emphasize that the H_2_O in this compound is enclathrated rather than being water of crystallization. Yet, signs of temperature-dependent dynamic interactions with the wall of the cages can be detected, and **1@**2H_2_O easily releases its water content even on standing and yields compound **2**. Surprisingly, hydrolysis products of **1** were observed even in the absence of aerial humidity, which suggests a unique solid-phase quasi-intramolecular hydrolysis. A mechanism involving successive substitution of the ammonia ligands by water molecules and ammonia release is proposed. An ESR study of the Cu-doped compound **2** (**2**#dotCu) showed that this complex consists of two different Cu^2+^(Zn^2+^) environments in the polymeric structure. Thermal decomposition of compounds **1** and **2** results in ZnMoO_4_ with similar specific surface area and morphology. The ZnMoO_4_ samples prepared from compounds **1** and **2** and compound **2** in itself are active photocatalysts in the degradation of Congo Red dye. IR, Raman, and UV studies on compounds **1@**2H_2_O and **2** are discussed in detail.

## 1. Introduction

The molybdates of the first-row 3D transition metals, including (non)-stoichiometric coordination polymers [1,2], are widely used as photocatalysts [3,4] and safe and high-density storage materials for the controlled release of ammonia in selective catalytic reduction of NO_x_ in exhaust gases or as fuel source for fuel cells [5]. Their synthesis with controlled composition, however, is a difficult task because their composition strongly depends on the synthetic conditions, as their layered structure permits building up various structural motifs and the formation of defector/substituted structures [6]. The M^I^M^II^(OH)MoO_4_ and M^I^(H)M^II^_2_(OH)_2_(MoO_4_)_2_ (M^I^ = Na, K, NH_4_^+^, M^II^ = Cu, Mg, Co, Ni, Zn) type compounds were first prepared by Pezerat [7] with variable compositions, including compounds containing ammonium and zinc ions (e.g., NH_4_Zn(OH)MoO_4_) [8].

Among these oxometallates, only limited information is available on tetraamminezinc(II) molybdate dihydrate ([Zn(NH_3_)_4_]MoO_4_**@**2H_2_O (Compound **1****@**2H_2_O) [9] (we use this unusual notation to express that, as it will turn out, the two water molecules do not occupy regular water-of-crystallization position in the crystal lattice). This salt is an unstable material, which, upon heating first releases 1 mol of NH_3_ and 0.6 mol of H_2_O at 50 °C, then 2 mol of ammonia and 1.3 mol of H_2_O at 120 °C, respectively. The loss of coordinated ammonia in the first decomposition step before completion of the elimination of water molecules is rather curious, because disruption of the coordination sphere should result in the collapse of the crystal lattice and so water from the crystallization should already be completely eliminated in the first decomposition step. To resolve this puzzle, we studied this process experimentally and performed quantum-chemical calculations to understand the order of elimination of water and ammonia ligands.

In this paper, we present our results on the formation of polymeric basic ammonium zinc molybdate (NH_4_)_x_(H)_1−x_Zn(OH)MoO_4_ with x = 0.92–0.94 (compound **2**) and show that this product forms from [Zn(NH_3_)_4_]MoO_4_**@**2H_2_O in solid-phase reaction routes either driven by external humidity or “self-hydrolysis” of compound **1****@**2H_2_O. Detailed spectroscopic and thermal analysis of the coordination polymer product and its precursor material, [Zn(NH_3_)_4_]MoO_4_**@**2H_2_O, are also discussed. The photocatalytic activity of the thermal decomposition product of the hydrated compound **1****@**2H_2_O and compound **2** has also been evaluated in detail.

## 2. Results and Discussion

### 2.1. Preparation of [Zn(NH_3_)_4_]MoO_4_**@**2H_2_O (Compound ***1*****@**2H_2_O) and NH_4_Zn(OH)MoO_4_ (Compound ***2***)

Tetraamminezinc(II) molybdate was prepared as dihydrate (**1****@**2H_2_O) by the method of Zelikman et al. [9] in a reaction of ZnMoO_4_ and stoichiometric aq. ammonia solution with subsequent cooling to 5–10 °C and drying under passing in NH_3_ stream. Another synthesis route for **1****@**2H_2_O was also designed, which starts from freshly prepared basic zinc molybdate as a precursor (formed from aq. sodium molybdate and aq. zinc sulfate in 1:1 molar ratio). The latter was dissolved in excess of the concentrated ammonia solution and **1****@**2H_2_O was precipitated by adding absolute ethanol. An excess of ethanol and ammonia is essential to prevent the hydrolysis of compound **1**. The product was always contaminated due to fast decomposition on drying. In open air, the amount of the decomposition product increased with increasing drying time (Figure 1) and was identified as compound **2**, (NH_4_)*_x_*(H)_1−*x*_Zn(OH)MoO_4_ (*x* = 0.92–0.94) (Card No.: 73-2389) with XRD and chemical analysis. To perform thermal, XRD, and IR studies and density determinations, compound **1** was precipitated as a powder from its aqueous solution with ethanol followed by washing the ethanol with diethyl ether. The thermal decomposition product of the hydrated compound **1** in air is ZnMoO_4_ [9] (see ESI Appendix A, Card. No.: 35-0765), with a mass loss of 31.2%, which corresponds to a loss of 2 moles of water in addition to all ammonia (the theoretical mass loss for the process is 31.62%). According to powder X-ray diffraction, the colorless microcrystalline compound **1** that formed has a cubic lattice with *a* = 10.345 Å (space group: F23 (no. 196)) (ESI Appendix A, ESI Appendix A). The cell volume is 1107.11 Å^3^, while the pycnometric density determined experimentally at 25 °C is 1.8612 g/cm^3^. The formula weight calculated for Z = 1 is *M* = 346 g/mol, and with the cell parameters, one obtains *d*_theor_ = 1.867 g/cm^3^. The difference of the pycnometric and theoretical densities corresponds to the presence of two molecules of water in the hydrated compound **1**. This stoichiometry is supported by the thermogravimetric measurement, presented later.

The dihydrate of compound **1** decomposes very easily on standing in the open air. Even in the absence of humid air, it slowly hydrolyzes, utilizing its own water content. Five days after preparation, the main final decomposition product was identified as (NH_4_)Zn(OH)(MoO_4_) (compound **2**, ESI Appendix A) in both cases. Section 3 below is devoted to the description of the similarities and differences in the mechanism of hydrolysis with an external or only internal water source.

### 2.2. Crystal Structure of Compounds ***1*** and ***2***

The structure of compound **1** has not yet been identified. Our efforts to grow single crystals to perform single-crystal X-ray diffraction measurements also remained unsuccessful. Compound **1** decomposes faster in its aqueous solutions even if cooled enough for the crystals to grow. It does not dissolve in non-polar organic solvents, whereas polar ones like DMSO or DMF or inorganic solvents like liquid NH_3_ or SO_2_ react with it [8]. Therefore, only powder XRD data could be used to solve the structure of compound **1**. To find appropriate starting coordinates, an analogous isostructural compound would be needed. We could not find compounds with known crystal structures of compositions [M(NH_3_)_4_]XO_4_·2H_2_O, [M(NH_3_)_4_(H_2_O)]XO_4_, or [M(NH_3_)_4_(H_2_O)_2_]XO_4_ that are isomorphous with the dihydrate of compound **1**. Surprisingly, the powder XRD patterns of some anhydrous tetraamminemetal tetraoxometallate compounds with monovalent XO_4_^-^ anions, e.g., [M(NH_3_)_4_](MnO_4_)_2_ (M = Zn or Cd) [10,11] or [Zn(NH_3_)_4_](ClO_4_)_2_ [12], were found to be analogous to those of the hydrated compound **1**. This is indeed surprising because neither the number of univalent negative anions nor their charge matches the composition of hydrated compound **1**. The structure of the permanganate and perchlorate compounds isostructural with **1****@**2H_2_O is unique in the sense that one perchlorate/permanganate anion is built into a network with the complex [M(NH_3_)_4_]^2+^ cations, making voids (O_4_N_4_(μ-H_12_)cube) that are occupied by the other perchlorate/permanganate anion.

Since **1****@**2H_2_O is isostructural with the salt of monovalent cations, in the crystal lattice of **1,** there is also a cation–anion network with cavities. However, since the cation:anion ratio is 1:1 in **1****@**2H_2_O, all molybdate ions are located in the cation–anion network, and there is no further anion in the compound that would, analogous to one of the permanganate ions, occupy the voids of the lattice. On the other hand, instead of the second anion, in **1****@**2H_2_O, there are two water molecules in the lattice that have no room in the cation–anion network, and it seems obvious that they are located in the cavities of the lattice, in the form of either two individual water molecules or a dimer.

In other words, a charged anion is substituted with two non-charged water molecules. Since the water molecules are embedded in lattice voids (Figure 2), they are neither coordinated nor are present as water of crystallization; instead, they are enclathrated (this is what the **@** symbol refers to in the abbreviation **1****@**2H_2_O). Since the water molecules located in the lattice voids are weakly bound, they can act the same way as water vapor [13]. An indication of the presence of some weakly bound components in the lattice of **1****@**2H_2_O is provided by the low-temperature DSC study of **1****@**2H_2_O, performed between −150 and 25 °C. Two endothermic effects have been detected at −103 and −17 °C with heats of 1.17 and 5.67 kJ/mol, respectively (ESI Appendix A). The temperature is very low, and the energy absorption is small, so it seems reasonable to assign these effects to some phase transitions and/or re-arrangement of the enclathrated water molecules. These might involve the change of the position of the two water molecules due to the change of their interaction with each other or with the atoms of the cage wall.

The crystal structure of compound **2** consists of sheets in which distorted ZnO_6_ octahedra are connected by tetrahedral molybdate groups [2]. The layers are stacked along the direction of the crystallographic c axis and are held together by hydrogen bonds. The ammonium ions are oriented through hydrogen bonds, and two different orientations for the ammonium ions are possible with equal probability. The ammonium ions cannot rotate freely. The zinc atoms in the structure are linked to each other through double OH bridges [2].

Further insight into the structure of **2** is provided by ESR spectroscopy. Since Zn and Mo are diamagnetic, we doped the compound with Cu^2+^ ions, which are expected to substitute Zn, and can provide information on its environment. The EPR spectrum of the powdered compound **2**#dotCu^2+^ formed this way suggests the presence of two different copper (and accordingly Zn) environments characterized by g^┴^ = 2.1135 and 2.2300 with A^┴^ 62.2 and 27.5 G, as well as g^║^ = 2.3252 and 2.0221 with A^║^ = 139.0 and 147.6 G. Surprisingly, one component has a “normal” (*g*∥ > *g*⊥) and the other an “inverse” (*g*⊥ > *g*∥) copper spectrum with a ratio of about 1:4. (ESI Appendix A). It seems reasonable to assume that, on one hand, the two different signals originate from the Cu^2+^(Zn^2+^) ions in the regular coordination environment [2], and on the other, from ions in environments distorted as a consequence of the ammonia deficiency.

### 2.3. Hydrolysis Reactions of Compound ***1***

As mentioned in Section 1, **1****@**2H_2_O easily hydrolyzes in the solid state. The hydrolysis takes place even when no external source of water is available, and in aqueous solutions as well. The main final product in both cases is compound **2**. Heating speeds up the hydrolysis in water. Upon boiling the aqueous solution of compound **1****@**2H_2_O for ≈ 10 min or 2 h in open air, the hydrolysis not only yielded compound **2**, but in addition, basic zinc carbonate (Zn_5_(OH)_6_(CO_3_)_2_) and ammonium heptamolybdate ((NH_4_)_6_Mo_7_O_24_) were also formed. Since compound **2** in an aqueous suspension is stable toward hydrolysis even on prolonged boiling, the latter two by-products must be formed from the hydrolysis of the starting compound **1**.

These experimental observations on the aqueous phase hydrolysis can be interpreted in terms of the following elementary steps [14,15]: (a) The tetraamminezinc cation releases some of its ammonia ligands; (b) the liberated ammonia is protonated with the formation of ammonium and hydroxide ions, making the pH basic; and (c) Zn^2+^ and MoO_4_^2−^ ions get into the solution. (d) The least soluble basic salts precipitate. These are {[NH_4_Zn(OH)(MoO_4_)]_2_}*_n_*, compound **2**, the main product, as well as Zn_5_(OH)_8_(CO_3_)_2_, which we identified as coming from the carbon dioxide absorbed by the alkaline solution. e) As a result of the removal of Zn^2+^ and the majority of the OH^−^ ions, only NH_4_^+^ and MoO_4_^2−^ ions remain available. The latter at the basic pH condensate into heptamolybdate ions known from the molybdate-polymolybdate equilibrium system [10], and then, together with the ammonium ions, they crystallize in the form of ammonium heptamolybdate (ESI Appendix A).

When **1****@**2H_2_O is left to stand in a closed, dry container, it slowly degrades and compound **2** is formed. This means that ammonia and water leave the lattice. As both are volatile, in principle, both could depart spontaneously. However, thermogravimetry shows that the rate of mass loss is negligible at temperatures below 80 °C. This suggests that the initial steps involve chemistry. This assumption seems to be confirmed by the XRD patterns taken after a day of reaction time, plotted in Figure 3.

One can see that in addition to the reactant **1****@**2H_2_O and product **2**, there are some intermediates in the system. Moreover, from the comparison of the top panels of Figure 3, it is also visible that the intermediates differ when external water is available compared to when the system is closed. Considering that the products are the same, and that the two water molecules located in the cavities of the lattice behave almost as if they were in the gas phase, it seems reasonable to assume that the reaction in the absence of an external water source is also hydrolysis, which is conducted by the lattice water molecules. To the best of our knowledge, this is the first example of a solid phase “quasi-intramolecular” self-hydrolysis reaction in a solid hydrate.

To understand the possible mechanism for the removal of ammonia ligands and the formation of ammonium ligands, we recall that the mechanism of a similar process has been determined in the case of anhydrous tetraamminecopper (II) molybdate [1]. It was shown that that compound, insoluble in water, undergoes a solid-state reaction when water molecules are available. The process was experimentally shown to start by NH_3_ release. The proposed mechanism is that the ammonia ligands are successively substituted by water molecules, and even the identity and structure of the intermediate mixed-ligand salts have been determined experimentally [1]. The final ammonia ligand was found to be so strongly bound to the transition metal ion that instead of leaving, it gets protonated by a water ligand coming originally from ambient humidity. It is likely that the mechanism for the decomposition of **1****@**2H_2_O in open, humid air is very similar, namely, stepwise formation of [Zn(NH_3_)*_n_*(H_2_O)_4-*n*_] complexes, where *n* decreases from 4 to 1 (compounds **3** (*n* = 3), **4** (*n* = 2), and **5** (*n* = 1)), and finally the last ammonia ligand in compound **5** is protonated by a water molecule in the lattice, yielding ammonium and hydroxide ions and with that, the formation of compound **2** (Figure 4).

In the absence of humidity, only the two water molecules located in the voids of the crystal lattice of **1****@**2H_2_O are available for the hydrolysis. Obviously, one of them is incorporated into the lattice of compound **2**. Since the H_2_O/Zn ratio is 2, no intermediates can primarily be formed with more than two H_2_O molecules per Zn. However, there are several cavities containing easily mobilizable water molecules in the close neighborhood of each coordinated Zn ion, and the constrained H_2_O/Zn ratio may make certain reaction channels more favorable and others less so. The basic mechanism is probably similar in the absence of external water molecules in comparison to when water is abundant.

### 2.4. Quantum Chemical Considerations on Self-Hydrolysis/Vapor-Phase Hydrolysis of Compound ***1***

As mentioned, the [Zn(NH_3_)_4_]^2+^ ions in **1****@**2H_2_O are located in the crystal lattice. The evolution of the compound in humid air or in a closed environment including any possible ligand exchange has to take place in the solid phase. Direct ligand exchange of the stronger bound ammonia ligands by water is not favorable. It seems rather reasonable to assume that the ligand exchange is initiated by coordination of a water ligand to the tetraamminezinc ions, and loss of an ammonia ligand if the energetics permits. The first step then is coordination of a water molecule coming from the environment or from the crystal lattice with the formation of the [Zn(NH_3_)_4_(H_2_O)]^2+^ (**6**^2+^) complex ion. If abundant water is available, hexacoordinated [Zn(NH_3_)_4_(H_2_O)_2_]^2+^ ions (cis and trans-**9**^2+^) can also be formed. These ions, however, do not initiate new reaction pathways because the products of water or ammonia release are the same pentacoordinated ions (**6**^2+^) formed by direct water addition.

To explore the possibilities for the further steps of hydrolysis, guided by the mechanism determined for the hydrolysis of the copper molybdate analog, we performed electronic structure calculations using density functional theory. Here we present the results obtained at the M05-2X/LANL2DZ level of theory. The purpose is to gain a qualitative picture of the structures and relative energies of the intermediates possibly involved in the reaction. We determined the structures and energies of all possible pure and mixed-ligand complexes of the Zn^2+^ ion from the mono- up to the hexacoordinated ones. Their most important structural parameters are listed in ESI Appendix A. The geometry of the ions is as follows: Hexacoordinated—distorted octahedral; pentacoordinated—trigonal bipyramidal (except Zn(NH_3_)(H_2_O)_4_, which has a metastable square pyramidal conformer); tetracoordinated—tetragonal; tris-coordinated—planar trigonal. In the pentacoordinated 1:4 mixed-ligand ions, the unique atom can be in axial or equatorial position, while in those with 3:2 ligand ratio, the two identical ligands can be both equatorial, both axial, or one equatorial, one axial. The relative energies of the conformers of the pentacoordinated complex ions with NH_3_:H_2_O ratios of 4:1, 3:2, 2:3, and 1:4 are also very small, being as little as 0.21 kcal/mol (for the NH_3_:H_2_O = 1:4), and not more than 4.1 kcal/mol (for NH_3_:H_2_O = 2:3). It seems to generally hold that in the most stable conformer, the ammonia ligand is in an equatorial position. We note that the absolute magnitude of the energy differences may be overestimated, but the tendencies/trends can be considered to be right. An indication that the calculated ligand binding energies are too high is that according to thermogravimetry, both water and ammonia loss can be observed at as low temperatures as 130 °C, while the calculated binding energies are in the order of several tens of kcal/mol. An additional complication is that our calculations are performed for the molecules in vacuo, while in the real system, the neighborhood of the ions certainly influences the energetics, but the magnitude of the effect is hard to estimate. For the purpose of orientation, the tendency of the calculated energy differences seems to be a reasonable starting point.

The addition of a water ligand to the [Zn(NH_3_)_4_]^2+^ ion is an exothermic reaction by about 27 kcal/mol. This supports the assumption that the initial step of the substitution of the ammonia ligands is water addition. The resulting complex can lose a water or ammonia ligand at the cost of 27 kcal/mol or 32 kcal/mol, respectively. Thus, the loss of an ammonia is much less favorable than that of a water ligand, but it is not excluded. The reactions are slow and can lead to equilibrium. The equilibrium can be shifted towards increasing ammonia to water substitution because once an ammonia molecule is liberated, it can leave the crystal lattice. Thus, similarly to the copper molybdate serving as a model for the mechanism, complexes with lower and lower ammonia content can be accumulated. As the number of water ligands increases in the penta-coordinated ions, the removal of any ligand becomes more and more endothermic, with each new water ligand at roughly 5 kcal/mol for the ammonia and 4 kcal/mol for the water loss. The tetracoordinated ligands can also lose a ligand, but the energy to be invested for that process is roughly 20 kcal/mol larger. The energy of ligand removal from tris-coordinated ions is again larger by roughly an additional 20 kcal/mol, and the gap between the energies required for ammonia and water loss increases. From the [Zn(NH_3_)(H_2_O)]^2+^ ion, the binding energy of the ammonia and water ligands is 115 and 93 kcal/mol, respectively, which suggests that, similarly to the copper molybdate, the removal of the last ammonia is not favorable. A possible network of elementary steps is displayed in Figure 4.

### 2.5. Vibrational and UV-VIS Spectra of Compounds ***1*****@**2H_2_O and ***2***

As mentioned, compound **1****@**2H_2_O consists of approximately tetrahedral Zn(NH_3_)_4_^2+^ cations with ZnN_4_ core, coordinated ammonia molecules (*C*_3v_), approximately tetrahedral MoO_4_^2−^ anions, and two enclathrated water molecules. Knowing the main crystal structural parameters, a correlation analysis could be performed (ESI Appendix A). The molybdate ion, ammonia, and Zn^2+^ ion were taken into consideration. Both the ZnN_4_ core and MoO_4_^2−^ ion are tetrahedral, thus the molecular, site, and factor groups are under *T*_d_ symmetry. There are four normal modes: Symmetric stretching vibration (*ν*_1_ or *ν*_s_); symmetric bending (*ν*_2_ or *δ*_s_); antisymmetric stretching (*ν*_3_ or *ν*_as_); and antisymmetric bending vibration (*ν*_4_ or *δ*_as_). The symmetric bending mode is doubly, while both antisymmetric modes are triply degenerate under *T*_d_.

The total number of factor group modes due to the internal MoO_4_^2–^ vibrations is four and corresponds to nine vibrational degrees of freedom, exactly as in the free anion. The hindered rotations and the hindered translations of the molybdate anions are both triply degenerate under *T*_d_ symmetry and remain so under both site and factor groups (ESI Appendix A). All four modes are Raman active, but only the *F*_2_ modes (*ν*_as_ and *δ*_as_) are IR active. The total number of factor-group modes due to the external MoO_4_^2–^ vibrations (hindered translations and hindered rotations) is six vibrational degrees of freedom: Three rotation-like motions and three translation-like motions (ESI Appendix A). The IR, far-IR, and Raman spectra of compound **1****@**2H_2_O can be seen in Figure 5 and ESI Appendix A. The IR and Raman data are given in ESI Appendix A. For the molybdate, ion band positions are *ν*_s_ > *ν*_as_, which represents an inversion of the normal behavior of MO_4_-type species (M = Cr, Mn, S, Se, Tc, Re) [16,17,18,19,20,21]. The relative positions of deformation bands for the MoO_4_^2−^ anion, *δ*_as_/*δ*_s_, may be >1, 1, or <1, and the intensities of symmetric modes are higher in the Raman than in the IR spectra [17].

The singlet/triplet nature and intensity relations of *ν*_s_(Mo-O) *ν*_as_(Mo-O) bands in the Raman (>>1) and IR (<<1) spectra allow unambiguous assignment of these modes. Based on these considerations, in compound **1**
**@**2H_2_O, unlike the assignment by Busey [17] for Na_2_MoO_4_ but in agreement with the result given by Clark [22] for ZnMoO_4_, the band positions are *ν*_s_(Mo-O) > *ν*_as_(Mo-O), and *δ*_as_ (Mo-O) > *δ*_s_(Mo-O) The *δ*_as_ band position is either mixed with or is partly overlapped with that of *ν*_as_(Zn-N). The *δ*_as_/*δ*_s_ > 1 situation for compound **1****@**2H_2_O and ZnMoO_4_ unambiguously show that the sensitivity of the symmetric and antisymmetric deformation mode positions depend on the chemical environment around the molybdate ion.

The appearance of the forbidden *ν*_s_(Mo-O) and *δ*_s_(Mo-O) modes in the IR spectrum of compound **1****@**2H_2_O confirms the distortion of the molybdate anion probably due to the hydrogen bond interactions with the ammonia hydrogens of the cation or the water molecules embedded in the cages. There is a rather large difference (~40 cm^−1^) between the center of gravity positions ($\sum_{i}^{n}h$_i_*ν*_i_/$\sum_{i}^{n}h$_i_) of modes in the IR and the Raman spectra. The reasons for such “splitting” are not clear at this moment.

The expected vibrational modes for the ZnN_4_ core of the complex cation are analogous to those for the molybdate anion. The correlation analysis of the internal ammonia ligand modes (*C*_3v_ molecular and site group, and *T*_d_ factor group) can be seen in Figure 6.

The total number of factor-group modes due to the internal NH_3_ vibrations is 10, which is equal to 24 vibrational degrees of freedom. The *F*_2_ modes are IR active, while *A*_1_, *E,* and *F*_2_ are Raman active. The F_1_ modes are inactive. The symmetric modes (*ν*_1_ or *ν*_s_, and *ν*_2_ or *δ*_s_) are singlets, while the antisymmetric modes (*ν*_3_ or *ν*_as_, and *ν*_4_ or *δ*_as_) are doubly degenerate vibrations under *C*_3v_. Regarding external modes of NH_3_ molecules, both *T*_xy_ and *R*_xy_ are doubly degenerate modes under *C*_3v_. The total number of factor-group modes due to the external vibrations equals 4 × 6 = 24 vibrational degrees of freedom (12 of rotational and 12 of vibrational origin). The *F*_2_ modes are both IR and Raman active. The *F*_1_ ones are inactive. *A*_1_ and *E* modes are only Raman active. The central zinc cation has one triply degenerate hindered translational mode, *T*_xyz_ (both site and factor group symmetries are *F*_2_).

In the assignment of cation modes in compound **1****@**2H_2_O, the IR and Raman data of the isostructural [Zn(NH_3_)_4_](MnO_4_)_2_, and the vibrational analysis results of the isotope-substituted Zn(^15^NH_3_)_4_^2+^ and Zn(^14^NH_3_)_4_^2+^ cations [23] were also considered. The assignments of bands belonging to the ZnN_4_ core and ammonia ligands are given in ESI Appendix A.

The appearance of the IR inactive *δ*_s_(Zn-N) mode confirms the distortion of *T*_d_ geometry around the ZnN_4_ core. Similarly, the IR inactive *A*_1_ and *E* modes (in *C*_3v_) of ammonia ligands are allowed, due to symmetry lowering induced by the evolution of extended hydrogen bond network involving the ammonia molecules and molybdate ions and the interaction of N-H bonds of the network with the water components in the cages. The bands at 1392 and 1301 cm^−1^ can be attributed to second-order transitions analogous to the first overtones of the *ρ*(NH_3_)(E) modes at ~700 and 650 cm^−1^.

The vibrational modes of enclathrated water in the spectra of compound **1****@**2H_2_O are expected to be similar to the monomeric or dimeric water modes in the gaseous state. The monomer water molecules must be disordered, as they occupy sites of *T*_d_ symmetry. The easiest (we found no better) way to account for this is to add three degrees of freedom for internal vibrations, three others for librations, and another three for hindered translations without employing the correlation method (Figure 7). An alternative approach would be to treat the water molecules as having spherical symmetry. The latter would imply a triply degenerate hindered translational mode *T*_xyz_, which is feasible, a triply degenerate librational mode *R*_xyz_, which is hardly acceptable, and a triply degenerate internal vibrational mode, which is unfeasible, leading to three additional modes of *F*_2_ symmetry (9 degrees of freedom).

The water molecules embedded in the cages may form various dimer structures [18], which may interact with the atoms in the cage wall. Basically, the O-H stretching modes are located on the left side of N-H bands on a wide shoulder. The band system around 1650 cm^−1^ may be attributed to the scissoring mode of water molecules. A librational mode of water appears at 905 cm^−1^ [17], rather high for almost free water. [17]. The changes in the bonding mode and motions of these water molecules can change some spectral features. The Raman spectra on cooling the compound **1****@**2H_2_O down to −180 °C (ESI Appendix A) showed better separation of combination/overtone bands in the region below 3000 cm^−1^. It suggests changes in the orientation/motion of freely rotating water molecules in the cages, which influences the hydrogen-bonded ammonia molecule’s band positions.

There are some second-order transition bands (analogous to combinations and overtones) around 2000 cm^−1^, which belong to the water bending mode with other low-wavenumber (<400 cm^−1^) modes [17], and the band system observed around 1060/1082 cm^−1^ might, again, belong to water modes related to some second-order transitions.

The IR and temperature-dependent Raman spectra of compounds **1****@**2H_2_O and **2** are given in Figure 5 and Figure 8 or ESI Appendix A. The Raman spectral features are independent of the temperature. A slightly better resolution could be found at −180 °C compared to at room temperature. A comparison of tetrahedral molybdate and ammonium ion vibrational modes with the data of NH_4_Cu(OH)MoO_4_ is given in ESI Appendix A.

The *ν*_s_(Mo-O) band could easily be assigned as an intensive singlet in the Raman spectra at various temperatures. The *ν*_as_(Mo-O) appears as a triplet, and the difference between the A_1_ and B_1_ components is Δ*ν* = 54 cm^−1^, lower than that observed in isotypic tetrahedral η^2^ -C_2v_ anions (Δ*ν* ∼ 95−124 cm^−1^) [1]. Thus, the molybdate ion is most likely not a chelating one in compound **2,** and the coordination polymer is built up from Zn-O-Mo units. The Zn-O-Mo and Zn-OH modes are assigned as a wideband system around 650 cm^−1^, whereas the rocking mode of water has appeared at 960 cm^−1^.

The presence of a combination band of the symmetric N-H deformation and a lattice vibration around 2000 cm^−1^ (*ν*_2_ + *ν*_6_) unambiguously shows the presence of the same type of hindered rotation as in the NH_4_Cu(OH)MoO_4_ [1], i.e., the ammonium is tied in place by relatively strong hydrogen bonds.

The compounds **1****@**2H_2_O and **2** do not absorb light in the visible light range, but as for other tetrahedral oxoanions, LMCT charge-transfer bands involving the MoO_4_^2−^ ions can be observed in the UV range [24,25,26]. Since the environment of the molybdate ions in compounds **1****@**2H_2_O and **2** is very similar, their UV spectra hardly differ (ESI Appendix A). Based on symmetry and Laporte’s rules, two LMCT transitions are permitted for the tetrahedral molybdate ion: 2*e* ← *t*_1_ and 3*t*_2_ ← *t*_1_ [25]. The higher-energy LMCT band at ~211 nm corresponds to the transfer of an oxygen 2p p electron into the antibonding 2*e* orbital whose major constituents are the d-orbitals of the Mo atom in MoO_4_^2−^ [25]. The second transition (3*t*_2_ ← *t*_1_) as in other solid molybdates [26] was found at ~260 nm in both compounds.

### 2.6. Thermal Behavior of Compounds ***1*****@**2H_2_O and ***2***

The thermal decomposition of the samples of compound **1****@**2H_2_O and compound **2** was conducted both and inert and oxidative atmosphere. The curves suggest that aerial oxygen does not significantly affect the decomposition mechanism (Figure 9A). Compound **1****@**2H_2_O decomposes at almost the same temperatures in air as in argon (the difference is less than 5 °C) and the mass of the solid residue at 400 °C is only ~1% less in air than in inert atmosphere. The overall mass loss during thermal decomposition of compound **1****@**2H_2_O up to 400 °C is 30.1% in argon and 31.2% in air, which are in good agreement with the theoretical loss (31.62%) of four ammonia and two water molecules from one dihydrate. In the temperature range from ~70 to ~200 °C, compound **1****@**2H_2_O decomposes in several overlapped processes with DTG maxima at 134, 142, and 160 °C. The evolution of enclathrated water overlaps with ammonia release. This suggests that the release of the weakly bound water becomes possible after partial disruption of the lattice. To gain better insight into the decomposition mechanism of **1****@**2H_2_O, online coupled TG-MS measurements were also done. The MS data (Figure 10) proved the simultaneous evolution of both lattice water (*m*/*z* 18 and 17) and ammonia (*m*/*z* 17 and 16) in the whole range up to 400 °C. The low-intensity peaks of fragments *m*/*z* = 14, 15, 28, 30, and 44 (N^+^, NH^+^, N_2_^+^, NO^+^ and N_2_O^+^) show that, due to a slight oxidizing ability of molybdate, a small amount of coordinated ammonia is oxidized by MoO_4_^2−^. Up to 220 °C, the overall mass loss is 27.4 and 28.4% in argon and air, respectively, and corresponds to the loss of two water and three ammonia molecules (26.45%) per a **1****@**2H_2_O unit. The mass losses are somewhat higher than the calculated value, because a small part of the oxygen content of the molybdate ions also leaves in the form of a small amount of ammonia oxidation products. The decomposition processes at about 250 °C in both inert and oxidative atmosphere are followed by an endothermic heat effect (ESI Appendix A).

The decomposition of compound **2** is a single-step process with DTG maximum at 245 and 246 °C in inert and oxidative atmosphere (Figure 9B). The reaction is endothermic. The molar decomposition heats were found to be 79.66 and 75.86 kJ/mol in air and argon, respectively. The thermal decomposition of compound **2** resulted in ZnMoO_4_ both in argon and air, and mass loss of 11.1% (argon) and 11.3% (air), which are somewhat less than the calculated mass decrease after loss of NH_4_^+^ and OH^−^ (13.46%). It means that the formula of the hydrolysis product (compound **2**) of compound **1** is close to [NH_4_Zn(OH)MoO_4_], but the “neutralization” of [(NH_4_)_x_(H)_1-x_Zn(OH)MoO_4_] is not complete (~88–90%).

TG-MS results show that only traces of molybdate reduction and ammonia oxidation (NO^+^, *m/z* = 30) appear in an inert atmosphere. The NO^+^ peak intensity is higher in air than in Ar, which can be attributed to a small extent of the oxidation of the released ammonia (ESI Appendix A). The intensity ratios of *m*/*z* = 17 and *m*/*z* = 18 (*m*/*z* = 17, NH_3_^+^ and OH^+^, *m*/*z* = 18, H_2_O^+^) in inert atmosphere show that ammonia and water formed together [27], whereas, in air atmosphere, the intensity of *m*/*z* = 16 peaks is high because of O^+^ (*m*/*z* = 16 from O_2_ fragmentation) presence.

### 2.7. Photocatalytic Activities of Compounds ***1*****@**2H_2_O, ***2*** and ZnMoO_4_ (Compound ***10***) in Organic Dye Degradation

Zinc molybdate can be used as a photocatalyst in the degradation of Victoria Blue [4] and other harmful dyes and chemicals like phenol [4] or nitrobenzene [3], and its photocatalytic efficiency strongly depends on its preparation route [7]. We tested the photocatalytic activity of compounds 1 and **2** as precursor molecules, and ZnMoO_4_ prepared from **1** at 220 (Compound **10**-220-**1**) and 350 (compound **10**-350-**1**) and from **2** at 350 °C (compound **10**-350-**2**). The test molecules were Congo Red and Methyl Orange (carcinogenic non-biodegradable direct azo dyes). The tests were done in UV irradiation in de-aerated aqueous solutions (2 × 10^−5^ M dye, 0.03 wt% catalyst, 375 nm wavelength) (ESI Appendix A). From the kinetic curves, we determined the reaction rate and the pseudo-first order rate coefficients by a linear fit to the −ln(A/A_0_)—t. The results obtained in the photocatalytic degradation of Methyl Orange and Congo Red dyes at 375 nm UV irradiation (18 W, pH = 5.6 and 5.7, respectively) are given in ESI Appendix A. We have found weak photocatalytic activity (~ca 2-fold acceleration) in methyl orange degradation in the presence of compounds **1**, **2,** and ZnMoO_4_ samples prepared at 350 °C from these precursors (ESI Appendix A).

The photocatalytic activity for oxidation of Congo Red, was found to be excellent, k_app_^.^10^−4^ increased from 1.0 to 22, 26, and 38 min^−1^ in the presence of compounds **1**, **2,** and the ZnMoO_4_ sample prepared at 220 °C (compound **10**-220-**1**), respectively (ESI Appendix A). The ZnMoO_4_ samples prepared at 350 °C showed minimal increasing/decreasing activity in photocatalytic degradation of Congo Red (k_app_^.^10^–4^ = 9 and 3 min^−1^, respectively) comparing that of with the blank sample (ESI Appendix A). Compound **1****@**2H_2_O is soluble in water and slowly hydrolyses, thus the photocatalytically active ingredient might be a dissolved species or formed in situ as well. The BET surface area of solid compound **2** and the ZnMoO_4_ samples (compounds **10**-220-1, **10**-350-1, **10**-350-2) were found to be 12, 27, 26, and 18 m^2^/g, respectively. As can be seen, neither the BET values (12–27 m^2^/g) nor the differences between morphologies (Figure 11a,b) explain the difference between the photocatalytic powers of these samples. Therefore, the differences in the photocatalytic activities might be attributed to the valence distribution of each species in the surface layer.

Based on the results mentioned above, the ZnMoO4 and NH_4_Zn(OH)MoO_4_ derived from compound 1**@**2H2O are promising, easy-to-prepare photocatalysts. They speed up the photodegradation of Congo red at 325 nm by more than an order of magnitude. Further XPS study of samples of compounds 10-220-1 and 10-350-1 is in progress to clarify the mechanism of photodegradation enhancement and test the technological power of these materials.

## 3. Materials and Methods

Chemical-grade reagents and precursors (ZnCl_2_.6H_2_O, MoO_3_, Na_2_MoO_4_, H_2_PtCl_6_, 0.1 M aq. NaOH, 0.1 M aq. HCl, 8-hydroxyquinoline, methanol, acetic acid, ammonium acetate, perchloric acid) were supplied by Deuton-X Ltd.

The zinc molybdate was prepared according to Zelikman [9] in the reaction of a stoichiometric amount of ZnO and MoO_3_ at 650 °C.
ZnO(s) + MoO_3_(s) = ZnMoO_4_(s)

To form the ammonia complex according to
ZnMoO_4_(s) + 4NH_3_(aq) = [Zn(NH_3_)_4_]MoO_4_,
ZnMoO_4_ was dissolved in 25% aq. ammonia solution at room temperature by intensive stirring. The undissolved residue was removed by filtration, and, by volume, 3 times more ethanol was added to the ammoniacal solution. The formed precipitate was filtered off immediately and washed with abs. ethanol and diethyl ether.

Compound **2** was prepared by contacting compound **1****@**2H_2_O with air for several days at room temperature.
Zn(NH_3_)4MoO_4_**@**2H_2_O(s) = NH_4_Zn(OH)MoO_4_(s) + H_2_O(g) + 3NH_3_(g)
The transformation and the end of the process was followed by powder XRD. The reaction time was found to be 1–5 days, depending on the humidity in the air. For elemental analysis, the samples of 1**@**2H_2_O and **2** were dissolved in dilute perchloric acid. The zinc content was measured as oxinate [28], while molybdate content was determined as PbMoO_4_ [28]. The ammonia content was determined by boiling it off with 0.1 M aq. NaOH and acidimetric titration and checked with gravimetry using H_2_PtCl_6_ as the precipitating agent [28,29].

Spectroscopical measurements (IR, far-IR, and Raman) were performed on Bruker Alpha FT-IR spectrometer (Bruker, Ettingen, Germany), a BioRad-Digilab FTS-60A spectrometer (Biorad, Austin, TX, USA), and a Horiba Jobin–Yvon LabRAM-type microspectrometer (Horiba Jobin Yvon Gmbh, Bensheim, Germany), respectively, and the details of measurements are given in our previous papers [20,28,30].

Thermoanalytical studies on compound **1****@**2H_2_O and compound **2** were performed on a TA Instruments SDT Q600 thermal analyzer coupled to the Hiden Analytical HPR-20/QIC mass spectrometer (TA ISntrument, New Castle, DE, USA and Hidden Co., Warrington, UK) and he Setaram LabsysEvo thermal analyzer (SETARAM, Lyon, France), respectively. DSC tests were performed with a Perkin Elmer DSC 7 instrument (Perkin-Elmer, Waltham, MA, USA). The details of the measurements are given in [22,29].

The PXRD and SEM meaurements were performed on a Philips Bragg Brentano parafocusing goniometer (Philips, Eindhoven, The Netherlands) and Zeiss EVO40 microscope (Carl Zeiss AG, Oberkochen, Germany) as descsribed in [31,32,33].

UV spectra was measured on a Jasco V-670 UV–Vis spectrometer (JASCO Co., Oklahoma City, OK, USA) as it was given previously [22,34]

Low-temperature (−196.15 °C) N_2_ adsorption measurements were performed after 24 h degassing at 110 °C on a NOVA 2000e (Quantachrome, Boynton Beach, FL, USA) automatic volumetric instrument. The apparent surface area *S*_BET_ was determined using the Brunauer–Emmett–Teller (BET) model [35]. Evaluation of the adsorption data was performed with the Quantachrome ASiQwin software (version 3.0).

CW-EPR spectra were recorded using a BRUKER EleXsys E500 X-band spectrometer (Bruker, Ettingen, Germany, microwave power 10mW, modulation amplitude 5G, modulation frequency 100 kHz). Measurements were performed at ambient temperature on 1 and 10% doped powder samples. As expected, for the 1% sample, since the average distance of the paramagnetic centers was significantly higher, better resolution spectra were obtained.

To evaluate the photocatalytic activity of the [Zn(NH_3_)_4_]MoO_4_ and its intermediates formed at 220 °C and 350 °C under air, as well as NH_4_Zn(OH)MoO_4_ and its decomposed product formed at 350 °C by isothermal heat under air, 1.0 mg of each compound was put into 3 mL of an aqueous solution of Methyl Orange (4 × 10^−5^ M) and Congo Red (2 × 10^−5^ M) dyes by using quartz cuvettes. The samples were left standing in dark for 1 h in order to reach the adsorption equilibrium. After that, they were submitted to a UV radiation source by Osram 18 W blacklight lamps (*λ* = maximum intensity at 375 nm). The cuvettes were placed 5 cm from each lamp and the absorbance was measured every 30 min during four hours by a Jasco V-550 UV-VIS spectroscope. The relative absorbance values of the most intensive peaks for Methyl Orange (464 nm) and Congo Red (497 nm) were considered to evaluate the catalysts’ activity in the degradation of dyes. The pseudo-first order reaction rate constants were determined by linear regression of the data considering the Lagergren model.

Density-functional theory was used with the M05-2X and CAM-B3LYP functionals and the LANL2DZ basis set containing pseudopotentials to describe the core electrons of the transitional metal atoms. We report the results of the M05-2X/LANL2DZ calculations as those produced by the other functional are very similar. For each optimized molecular geometry, vibrational frequency analysis was performed to confirm that the stationary points are minimal. The discussion of energetics is based on the classical energy levels because the structures are rather fluxional and the determination of even the zero-point energies (zpe) is incorrect with the harmonic approximation. The simplest anharmonic correction, the hindered rotation approximation, is not of help because the most important factor is not the hindering of the NH_3_ and H_2_O rotors but Berry’s pseudorotation of the complex ion. All calculations were performed with the Gaussian 09 suite of programs [36].

## 4. Conclusions

A detailed analysis of the structure and properties of tetraamminezinc molybdate supplemented by two water molecules, **1****@**2H_2_O, of its conversion to basic ammonium zinc molybdate, **2** as well as those of the product **2** have been performed. The crystal structure of **1****@**2H_2_O, hitherto unknown, has been determined by finding isostructural analogs and powder XRD, since single crystals could not be prepared because the hydrolysis of the compound is faster than the crystal growth. There is a network a tetramminezinc cations and molybdate anions surrounding cavities occupied by the water molecules. Although the cage wall contains electronegative O and N atoms, the interaction between the caged water molecules and the wall proved to be weak, possibly due to the symmetry of the cage in which none of the positions are favorable. Thus, the water molecules do not behave as water of crystallization, and their state is close to those in clathrates. Their mobility is reflected in numerous properties such as vibrational spectra or DSC results. Detailed assignment of the IR/Raman spectra and interpretation of thermal decomposition of the compound have been provided. The compound 1**@**2H_2_O in aqueous solution is converted relatively fast to compound **2** (10 min boiling) and is contaminated by basic zinc carbonate and ammonium heptamolybdate. Rather interesting is the solid-phase hydrolysis of **1****@**2H_2_O in the presence, and especially in the absence, of aerial humidity. The probable mechanism for this non-trivial reaction is analogous to that determined for the close analog of compound 1**@**2H_2_O, tetraamminecopper molybdate; namely, successive substitution of ammonia ligands by water. The feasibility of this mechanism for **1****@**2H_2_O is supported by electronic structure calculations using density functional theory. The curiosity of the hydrolysis of **1****@**2H_2_O in a closed system is that the hydrolyzing agent in the solid-phase reaction comes from the lattice itself, and the reaction can be classified as “quasi intramolecular self-hydrolysis”. The thermal decomposition of both **1****@**2H_2_O and **2** yields zinc molybdate but with different nanostructure. Both were found to be photocatalysts and their activity was characterized.

Interesting similarities and differences can be revealed when comparing the structural information on **1****@**2H_2_O and compounds with closely related constitution, such as tetraaminecopper molybdate, tetraamminezinc permanganate, and -perchlorate. The skeletal structure of these salts is the same, but in the compound **1****@**2H_2_O, due to the charge difference between the MO_4_^−^/MO_4_^2−^ ions, there is no need for a charged particle to compensate the charge of the cation, which in the crystal lattice of the salts with MO_4_ occupies the voids, therefore a neutral species, namely water, occupies the cavities. This results in curious structural features such as clathrating in a cavity and unusual properties such as solid phase self-hydrolysis. A possible method for accurate structure determination is offered by the fact that a similar compound, tetraamminezinc (II) tungstate, also contains water [37]. We are planning to study the analogous tungstate and other compounds with the purpose of finding a relatively stable material whose structure could be solved by single crystal X-ray diffractometry and use the results as a basis in Rietveld refinement of the structure of the molybdate.

## Figures and Tables

**Figure 1 molecules-26-04022-f001:**
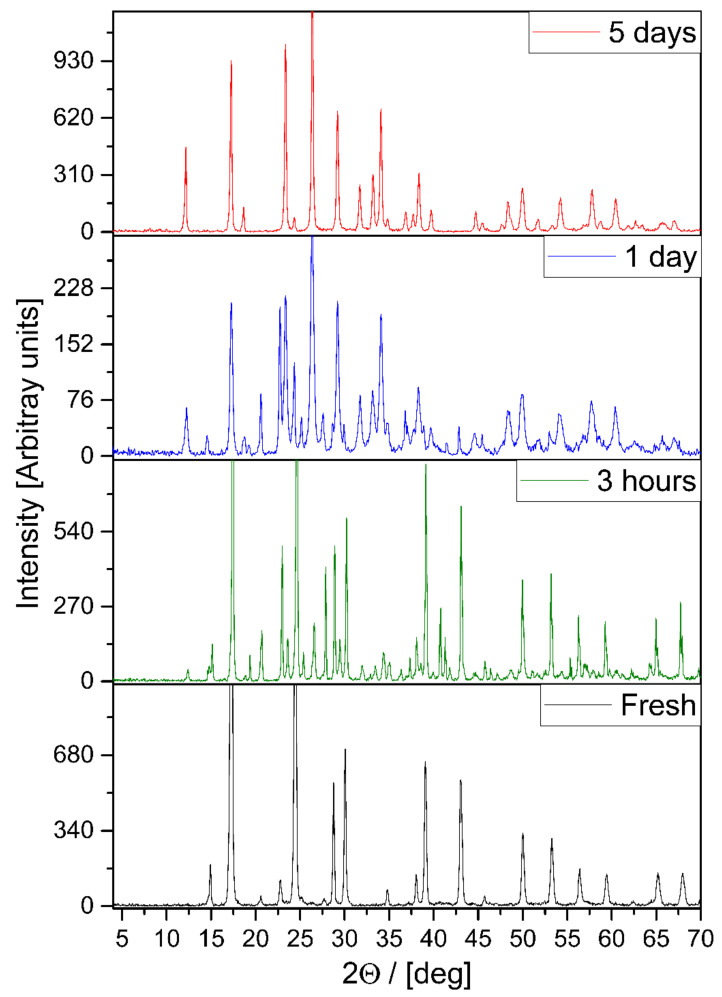
XRD of compound **1** and its decomposition products on standing in the open air.

**Figure 2 molecules-26-04022-f002:**
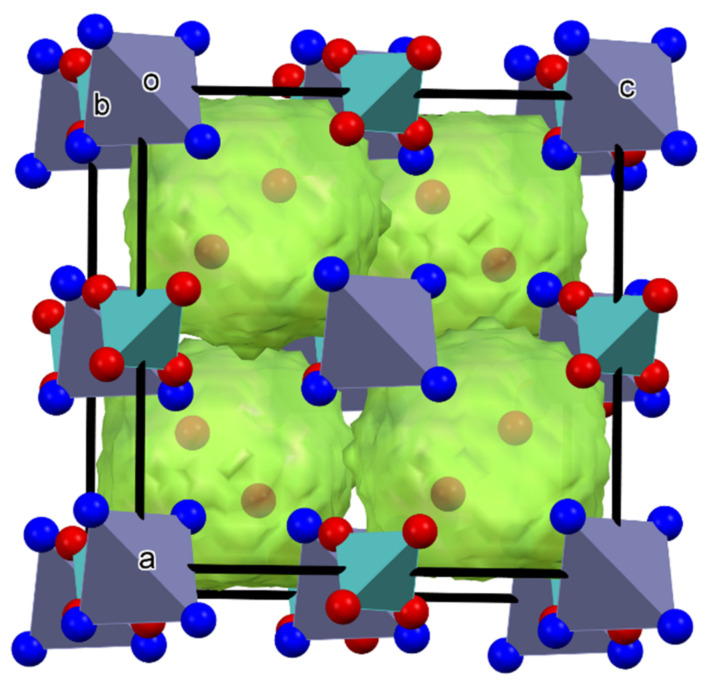
Voids for enclathrated water molecules (water dimer) in the lattice of compound **1****@**2H_2_O his is a figure. Schemes follow the same formatting. (a,b,c-axes direction, o-origo).

**Figure 3 molecules-26-04022-f003:**
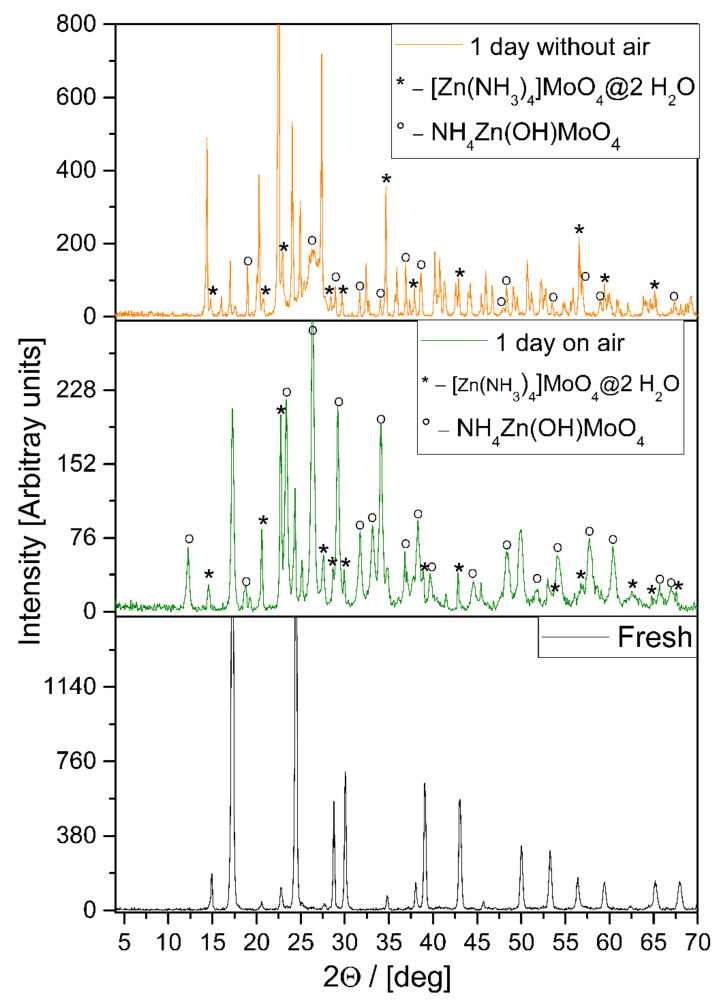
Comparison of the XRDs of decomposition intermediates formed from **1****@**2H_2_O in open air and in a closed system.

**Figure 4 molecules-26-04022-f004:**
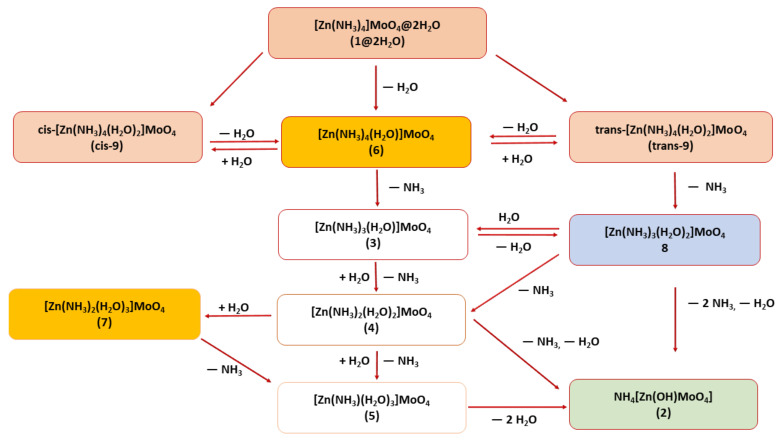
Possible pathways of compound **2** formation by self-hydrolysis/water vapor-mediated hydrolysis of compound **1**@****2H_2_O (red boxes contain the sources of the tetraammin-monoaqua salt; yellow and blue indicate the penta, whereas colorless boxes the tetracoordinated intermediates; green represents the final decomposition product).

**Figure 5 molecules-26-04022-f005:**
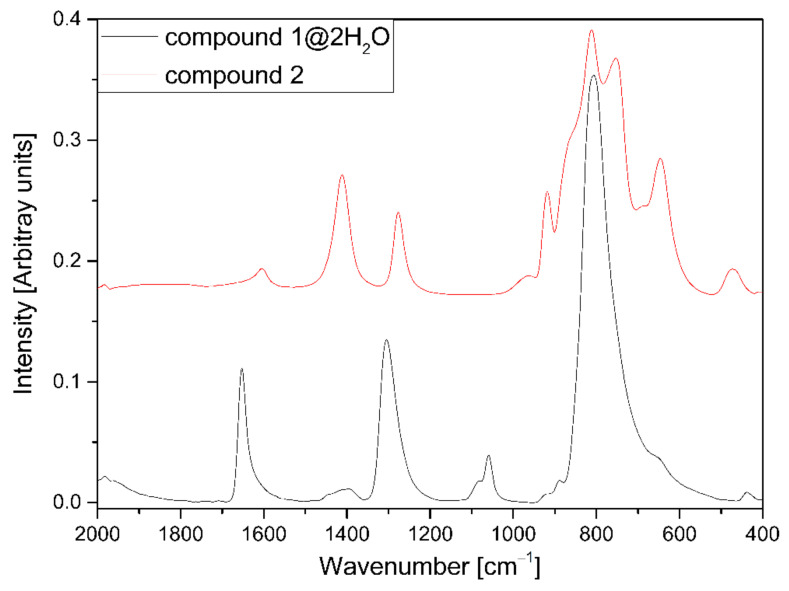
IR spectra of compound **1****@**2H_2_O and **2** between 2000 and 400 cm^−1^.

**Figure 6 molecules-26-04022-f006:**
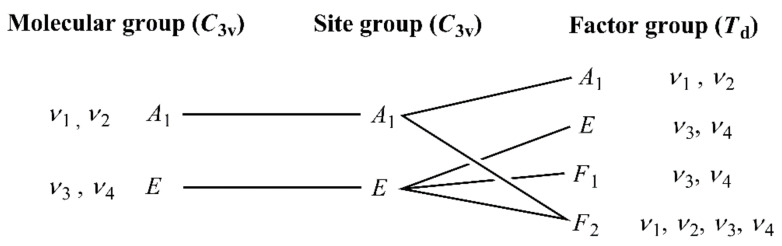
Correlation analysis of internal ammonia modes in compound **1***2H_2_O.

**Figure 7 molecules-26-04022-f007:**
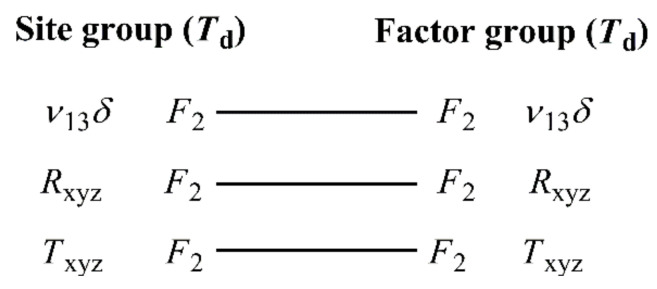
Correlation analysis for water molecules in compound **1****@**2H_2_O.

**Figure 8 molecules-26-04022-f008:**
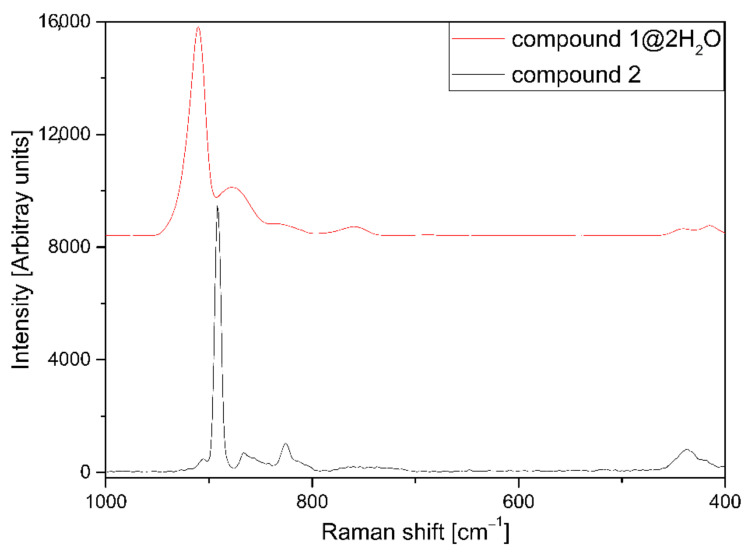
Raman spectra of compound **1****@**2H_2_O and **2** between 1000 and 400 cm^−1^ at −180 °C.

**Figure 9 molecules-26-04022-f009:**
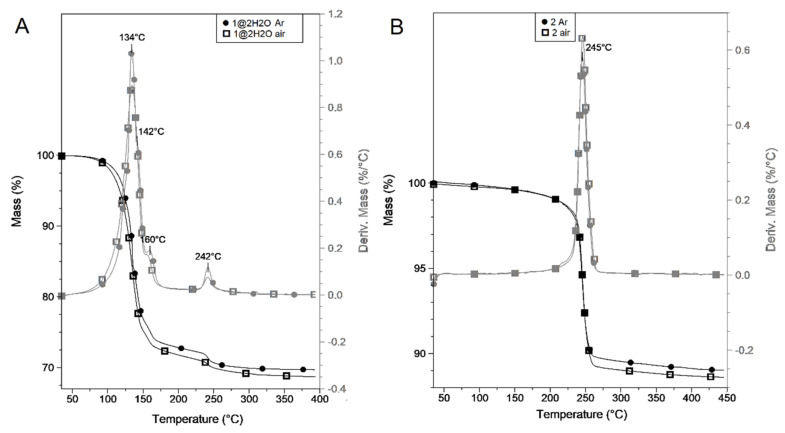
TG/DTG curves of compound **1** (**A**) and **2** (**B**) in argon and air.

**Figure 10 molecules-26-04022-f010:**
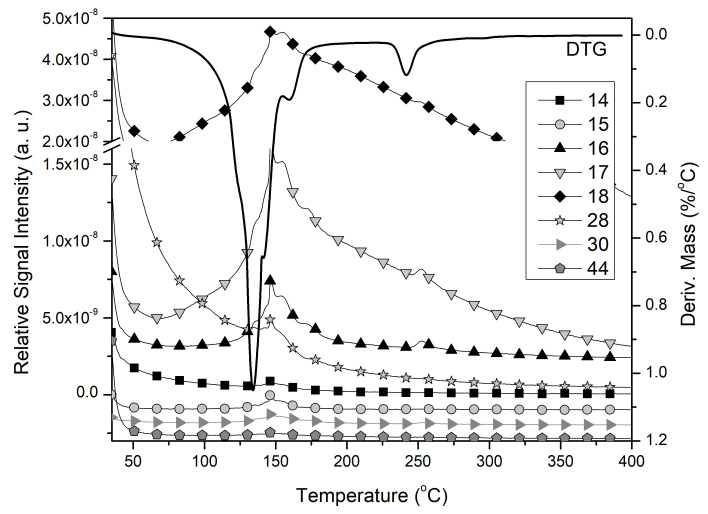
TG-MS curves of compound **1** in argon.

**Figure 11 molecules-26-04022-f011:**
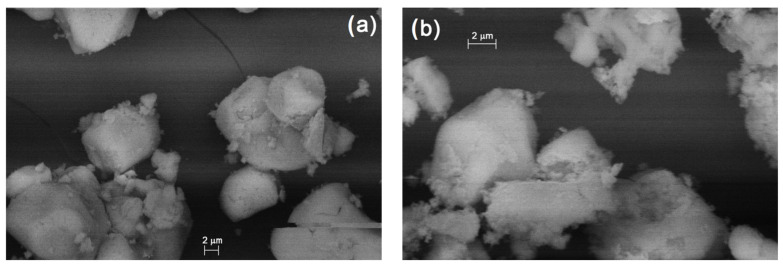
Morphology (SEM) of ZnMoO_4_ samples prepared at (**a**) 350 °C from compound 1**@**2H_2_O (compound 10-350-1) and (**b**) compound 2 (Compound 10-350-2).

## Data Availability

Data sharing is not applicable to this article. The data presented in this study are available on request from the corresponding author.

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
