# Peer review of "Solid-Phase “Self-Hydrolysis” of [Zn(NH3)4MoO4@2H2O] Involving Enclathrated Water—An Easy Route to a Layered Basic Ammonium Zinc Molybdate Coordination Polymer"

_molecules, 2021, doi:10.3390/molecules26134022_

Round 1
Reviewer 1 Report
The manuscript “Enclathrated water and its role in the solid phase "self-hydrolysis" of [Zn(NH3)4MoO4@2H2O] — an easy route to a layered 3 basic ammonium zinc molybdate coordination polymer” is well written and provides enough experimental evidence for the suggested conclusions. The solid state transformation of the original phase is well analyzed and might be of interest to the readers. I don’t have any specific corrections to propose.
Author Response
Reviewer 1. I don’t have any specific corrections to propose.
Answers: The reviewer did not ask any correction.
Reviewer 2 Report
The paper reports on the preparation of a water-enclathrated compound, NH3-Zn(II)-MoO4, and its self-hydrolyzed product, NH4Zn(OH)MoO4. The structures of reported compounds are characterized by PXRD, EPR, and thermal studies. While, the photocatalytic activities for degradation of organic dye of the reported compounds are also reported. The studies in the structure and pathway of hydrolysis of the presented compounds have been relatively well performed and are well described in this paper. As a conclusion, the manuscript appears to be appealing for the scientific community and could be recommended for publication. However, before doing so I request the authors to address the following minor revisions.
- It is suggested to use the consistent abbreviations for “compound 1” and “compound 1@2H2O”, if the enclathrated water molecules are important in compound 1@2H2
- 3, L. 107-108. The authors mentions that the compound 1@2H2O decomposes and hydrolyses to compound 2 very easily on standing in the open air through self-hydrolysis by the enclathrated water. However, the evidence of the role of enclathrated water in the structure changes from compound 1@2H2O to compound 2 is not clear. Could this self-hydrolysis can be prevented if the enclathrated water was fast removed by thermal treatment? The temperature dependence of in-situ PXRD analysis may provide this information.
- It was suggested to add the detail of preparation for compound 1@2H2O and compound 2 to experimental section.
- A simple synthetic scheme of compound 1@2H2O and compound 2 were suggested to added.
After corrections of these points I will be happy to see the paper published in Molecules.
Author Response
Reviewer 2.
- It is suggested to use the consistent abbreviations for “compound 1” and “compound 1@2H2O”, if the enclathrated water molecules are important in compound 1@2H2O
Answer: We have revised all occurrences of “compound 1” to “compound 1@2H2O”.
- 3, L. 107-108. The authors mentions that the compound 1@2H2O decomposes and hydrolyses to compound 2 very easily on standing in the open air through self-hydrolysis by the enclathrated water. However, the evidence of the role of enclathrated water in the structure changes from compound 1@2H2O to compound 2 is not clear. Could this self-hydrolysis can be prevented if the enclathrated water was fast removed by thermal treatment? The temperature dependence of in-situ PXRD analysis may provide this information.
Asnswer: The compound 1@2H2O hydrolyzes both with the external humidity or in the lack of that “using” its own water content. Fast heating of compound 1@2H2O cannot be used to quickly eliminate the loosely bound water, because during the thermal treatment, , ammonia and water evolve together. The reason for this is that heating increases the rate of hydrolysis.
- It was suggested to add the detail of preparation for compound 1@2H2O and compound 2 to experimental section.
Answer: It has been added.
- A simple synthetic scheme of compound 1@2H2O and compound 2 were suggested to added.
Answer: It has been added.
Reviewer 3 Report
The manuscript describes the synthesis and in-depth characterization of a basic ammonium-Zn-molybdate polymer with encapsulated (clathrated) water molecules. While single crystals could not be obtained due to the relatively rapid decomposition of the material, its structure, the positions of the water molecules, and its (and related compounds') photocatalytic activites have been investigated using a range of experimental and theoretical techniques. The work should be of interest to a broad inroganic research community. The results are clearly presented and support the authors' conclusions. I recommend publication in Molecules after the following minor points have been addressed:
1) lines 149 and 150: please make sure that there is no break between the negative sign and 150.
2) line 163: 'paramagnetic' should be 'diamagnetic'
3) lines 170-173 and 499-508: the text has a smaller font size than the rest of the manuscript.
4) line 520: please provide the reference to the Zelikman paper that was followed during the synthesis.
5) The authors refer to the supporting information, I have not been able to find it.
Author Response
Reviewer 3.
1) lines 149 and 150: please make sure that there is no break between the negative sign and 150.
Answer: It has been corrected.
2) line 163: 'paramagnetic' should be 'diamagnetic'
Answer: It has been corrected.
3) lines 170-173 and 499-508: the text has a smaller font size than the rest of the manuscript.
Answer: It has been corrected.
4) line 520: please provide the reference to the Zelikman paper that was followed during the synthesis.
Answer: It has been corrected.
5) The authors refer to the supporting information, I have not been able to find it.
Answer: It has been added.